# A Study on Deriving Improvements through User Recognition Analysis of Artificial Intelligence Speakers

**Seong-Jeong Yoon *** and **Min-Yong Kim ***

School of Business, Kyung Hee University, 26 Kyungheedae-ro, Dongdaemun-gu, Seoul 02447, Korea
* Correspondence: sj9416@naver.com (S.-J.Y.); andy@khu.ac.kr (M.-Y.K.); Tel.: +82-2-961-2147 (M.-Y.K.)

**Featured Application: People think that artificial intelligence speakers are not speakers that contain artificial intelligence. We would like to suggest improvement points through a perception survey on why this is the case.**

**Abstract:** Recently, artificial intelligence speakers have been used a lot in homes and offices. However, users say that it is an automated speaker, not an artificial intelligence speaker. Regression analysis was performed by applying the Value-Based Acceptance Model (VAM) to see if there are any improvements to the negative perceptions of users mentioned above. As a result of the regression analysis, improvements were needed for convenience and security threats, and it did not reach the level of anthropomorphism such as with humans. In addition, it is concluded that the factors that positively affect the perceived value are usefulness and enjoyment and that they are somewhat satisfied with the burden of technical difficulties, cost, and reliability of the information. In conclusion, artificial intelligence should continuously collect various data and provide information or suggest choices and alternatives through the process of analysis, learning, and inference. However, as a result of this study, it is concluded that it is similar to an automated machine that simply finds the data among many data connected to the Internet, plays music, and connects to a site where you can shop and process it non-face-to-face. The rationale for being similar to an automated machine is that it has not reached the level of anthropomorphism.

**Keywords:** AI speaker; user's recognition; value-based acceptance model; anthropomorphism; perceived value; information reliability

## 1. Introduction

Artificial intelligence speakers are often used in our daily life. The purpose of this study is to measure satisfaction with this and the intention to continue using it. People around me use AI speakers and do not think they are AI speakers. The reason for this is as follows. First, voice recognition is not good in terms of ease. Second, in terms of anthropomorphism or intelligence, it is not about analyzing (big) data and telling an intelligent story through repeated learning of data but about finding and telling a vast amount of data. Third, it is not yet clear what usefulness the user expects for the user value. However, it is all about talking with voice recognition and automating functions according to time settings. Fourth, if you ask a question to an artificial intelligence speaker, there is no reliability in the answer. Too often, they fail to provide the relevant information for the question. Fifth, the elements that artificial intelligence speakers give pleasure are provided in extremely limited conversations. Music, video, and conversation are everything. The limiting factor in providing enjoyment is illustrated by the example of music. If you tell the title of the music, it will be heard as it is. The AI speaker should be at a level that can play music according to the user's emotional state and surrounding environment, but the current level is not that much. The music is not played through recognition of the surrounding environment, such as the current location, temperature, humidity, and the user's mood

and weather. In conclusion, recognition used as a marketing tool for telecommunication companies that finds vast amounts of data as a kind of speech recognition machine is not artificial intelligence. Then, as mentioned above, we would like to reveal whether it is just an automated speaker or an intelligent speaker for users who have used artificial intelligence speakers directly and often. Through this, future artificial intelligence speakers launched by each telecommunication company on the market will. We would like to suggest what needs to be developed by focusing on what. Without suggesting this, it is self-evident that products will come to the market by replacing only the shape of the so-called artificial intelligence speaker. Ultimately, it is necessary to measure the sustainable use of artificial intelligence speakers in the future.

## 2. Theoretical Background

This paper intends to conduct a causal relationship analysis using the Value-Based Acceptance Model (VAM) in order to identify the intention of recommending using artificial intelligence speakers. Opinions about artificial intelligence speakers currently on the market are diverse, including negative aspects rather than positive ones. The only measurement model that can measure this is VAM. The reason for using the VAM model in this study is to supplement the blind spots found in the following existing studies. In the study of Lee et al. (2020), positive factors such as anthropomorphism, service diversity, and ease of use were set as independent variables using the technology acceptance model, and perceived invasion of privacy was set as negative factors [1]. A causal relationship analysis was performed on the parameters, including positively perceived usefulness and perceived pleasure. As a result of the analysis, it is concluded without any particular issue that perceived pleasure affects intention to use rather than perceived usefulness. These results do not reflect the problems of the current technical level of artificial intelligence at all. In the study of Yoo et al. (2020), the usefulness of artificial intelligence speakers for the elderly living alone was analyzed using a technology acceptance model to analyze the causal relationship [2]. The independent variables were set as perceived usefulness, excluding negative factors, quality of life, and perceived ease of use. Use attitude was measured as a parameter, and intention to use as a dependent variable. It is also simply concluded that positive independent variables have a positive effect on the parameters and dependent variables. There is no antecedent variable for anthropomorphism, a characteristic of artificial intelligence. Rather than measuring the causal relationship targeting artificial intelligence speakers, this has no meaning except for measuring the technology acceptance of general smart devices. Therefore, in this study, improvement points should be suggested after setting and measuring the anthropomorphism of the antecedent variable and positive and negative factors. The model to be applied measures the independent variable in two groups. In other words, they are the constructs in the aspect of Benefit and the constructs in the aspect of Sacrifice. The purpose of this study is to determine whether independent variables have a positive effect on perceived value first and, finally, to analyze whether they affect the intention of recommendation.

### 2.1. Types of Artificial Intelligence on the Market

The oldest AI speakers are Xiomi's Mi AI Speaker, released in July 2007, and Alibaba's Tmall Genie. By 2022, 15 years have passed. In Korea, it is a product called Nugu developed by SKT, which was released in September 2016. After 2016, many telecommunication companies and home appliance manufacturers in Korea started to release artificial intelligence speakers. Its main functions are online shopping, listening to music, and searching. The most expensive AI speaker abroad is Apple's Siri with a price of USD 349, and LG Electronics' Naver Clova is the most expensive in Korea with a price of USD 400. The types of artificial intelligence speakers released on the market are as follows [3,4].

However, in order to bear the high cost and to popularize the market, telecommunication companies have changed the method of charging the telecommunication usage cost as an additional service concept [4].

In Korea, many artificial intelligence speakers have been used since 2016. What are people's expectations and perceptions about artificial intelligence speakers that have been used for about 6 years? Do artificial intelligence speakers have the characteristics of artificial intelligence? Do artificial intelligence speakers that are released on the market with similar functions and different shapes provide users with artificial intelligence functions through the functions of reasoning, judging, and learning (deep learning or machine learning) [5–7]? Or is it simply adding a large amount of data to a few functions to build a diverse database and release it for market launch as if it were learned? Before conducting this study, I heard that it is similar to an automated machine that collects a large amount of data from many people and gives an appropriate answer [8]. In particular, do users think that technology has evolved over the course of 15 years for the overseas market launch and 6 years for the domestic market? In response to these questions, we want to ask users about it and propose improvements.

In the study by Kim et al. (2022), the market growth rate of artificial intelligence speakers is defined as follows. Table 1 describes the market growth potential of artificial intelligence speakers as follows. The vendors with the largest market share are Amazon (26.60%), Google (17.30%), and Baidu (15.60%) in that order, with similar growth rates. However, Apple (7.90%) is the second fastest growing company after Amazon. As shown in the table below, those belonging to other categories are those of Korea in Figure 1. It was found that the growth rate was only 0.20%. The growth rate of artificial intelligence speakers in the global market may be slightly different, but it is clear that they are growing. However, what this study intends to focus on is whether such growth has simply grown as a marketing strategy despite the lack of technological level [8].

**Table 1.** Artifical Artificial intelligence speaker market forecast by [9].

| Vendors | Market Share (2021) | * Shipments (2020) | * Shipments (2021) | Growth Rate |
|---------|---------------------|--------------------|--------------------|-------------|
| Amazon | 26.60% | 33.60 | 42.40 | 8.80% |
| Google | 17.30% | 23.80 | 27.60 | 3.80% |
| Baidu | 15.60% | 19.40 | 24.80 | 5.40% |
| Alibaba | 12.60% | 17.10 | 20.00 | 2.90% |
| Apple | 9.60% | 7.30 | 15.20 | 7.90% |
| Xiaomi | 6.30% | 10.60 | 10.00 | −0.60% |
| Others | 12.00% | 18.90 | 19.10 | 0.20% |
| Total | 100.00% | 130.70 | 159.10 | 28.40% |

* Shipments: shipments in millions of units.

### 2.2. Benefit Factor

This study intends to measure by adding anthropomorphism [9,10] to convenience, usefulness, and pleasure among the constructs mainly used as positive factors suggested by the existing Value-based Acceptance Model (VAM) [11–14]. Considering that the AI speaker was launched in 2007 by Xiomi's Xiomi AI Speaker, we want to measure whether there are characteristics similar to those of a person who can represent the characteristics of artificial intelligence due to technological progress. In addition, technical difficulties, cost burden, and security threats as negative factors will be measured as well as the reliability of the information.

In terms of the benefits of artificial intelligence speakers, we want to apply the paradigm of CASA (Computers as Social Actors [15]) rather than simply measuring the perceived usefulness or the perceived ease of use. CASA recognizes the relationship between computers and humans as a social relationship. In other words, it is possible to recognize the AI speaker as a human rather than a machine and feel emotional sympathy.

Such emotional communion includes perceived anthropomorphism, intimacy, reliability, and perceived enjoyment similar to humans [16].

| Foreign products | | | | | |
|---|---|---|---|---|---|
| Manufacturer | Amazon | Google | Apple | Alibaba | Xiomi |
| Product name | Echo | Home | HomePod | Tmall Genie | Mi AI Speaker |
| AI Platform | Alexa | Google Assistant | Siri | AliGenie | XioAi |
| Price | $180 | $129 | $349 | $40 | $45 |
| Release Date | 14, November | 16, November | 17, October | 07, July | 07, July |
| Characteristic | Buy online through Amazon | Google searchable, big data available | Apple Music available. Enhanced speaker + security function | Shopping function emphasis | - |

| Domestic products | | | | | |
|---|---|---|---|---|---|
| Manufacturer | KT | SKT | NAVER | KAKAO | LG Electronics | SAMSUNG Electronics |
| Product name | Giga Gini | Nugu | Wave | KakaoMini | ThinkHub | GalaxyHome |
| AI Platform | Giga Gini | Nugu | Clova | Kakao i | Clova(Naver) | Bixby |
| Price | $300 | $150 | $150 | $120 | $400 | $300 |
| Release Date | 17, January | 16, September | 17, August | 17, November | 17, November | 18, August |
| Characteristic | speaker and set-top box | Music playback, shopping, delivery | Music playback, Naver search, translation | Music playback, Daum search, messenger | Music playback, Naver search, translation | Emphasis on speakers based on Harman sound technology |

**Figure 1.** Types of domestic and foreign AI speakers [3,4].

Perceived anthropomorphism is when non-human animals, certain products, computers, and robots are given certain human-like attributes and emotions. Then, such things and objects think and interact similar to humans [17–19]. However, as a result of interviewing a person who uses an artificial intelligence speaker in advance, they say that it does not seem to be at that level. Existing research on AI speaking is as follows. A study by Pipitone et al. (2021) suggests that artificial intelligence speakers should be able to simulate the human inner world. Here, it is pointed out that the simulation should be able to speak subjectively rather than simply talking only the inputted sentences. In addition, it should be possible to use the informal language generated by artificial intelligence as well as the official language [19].

This is called anthropomorphism. However, the AI speaker investigated in this study does not have such a function. The date of this study is 2022. However, the artificial intelligence suggested by Pipitone's research compared to the definition of speaking anthropomorphism, AI speakers in products on the market are not AI at all.

As for the convenience of artificial intelligence, the following examples are given. The AI speaker is placed next to the TV to operate the TV or it is used for Internet browsing. We are using a service that plays music to a dog when it is alone in the house through an artificial intelligence speaker or provides food according to the time of day [20]. As for the usefulness of artificial intelligence, the following examples are given. Artificial intelligence speakers are included in navigation systems when driving a vehicle and they are used to generate the shortest route by recognizing the destination by voice. In addition, it can be seen that an artificial intelligence speaker was used as a conversation partner for the elderly [21]. The intimacy created by using artificial intelligence speakers is said to provide psychological help to the elderly living alone, people living alone, and the socially disadvantaged [21]. Intimacy is said to be an important factor influencing the

formation and maintenance of interpersonal communication. The intimacy here is that it is possible to form intimacy between the AI speaker and people rather than interpersonal communication. In social psychology, it is said that this kind of intimacy provides people with satisfaction, love, and trust in interpersonal relationships [22].

The expectation that perceived enjoyment will include all entertainment functions, including dialogue, music play, and information on culture, art, and movies, is not realistic [23]. Existing studies can also examine the pleasure of using the function of singing a song after some elderly people ask the AI speaker to play music. In this study, considering the level of enjoyment provided by the current artificial intelligence speaker, we intend to limit the enjoyment of the conversation itself. Here, the dialogue itself refers to the enjoyment of the subject matter and content. There are many studies on the hedonistic value in the existing studies as well [23].

*2.3. Sacrifice Factor*

The sacrifice factors of artificial intelligence speakers can be presented as the technicality, burden of cost, threat to security, and distrust of information [24,25]. First, technicality measures the user's experience with technical difficulties. Among the technical difficulties of artificial intelligence speakers, the most direct factor is voice recognition. Recently, deep neural network classifiers have been developed to improve speech recognition [26,27]. Artificial intelligence should be able to listen to the information provided when a user makes a query and make the wisest decision while reducing time and cost. Nevertheless, the reality is that current artificial intelligence speakers do not go beyond the level of accessing and reading the data in response to a user's query [27].

The burden of cost will be a flat-rate payment in the form of a purchase or a monthly payment. As a result of self-investigation for this study, Alibaba's Tmall Geni is the cheapest with a price of USD 40, and LG electronics' ThinkHub is the most expensive with a price of USD 400. Recently, in order to reduce this cost burden, KT has offered the "GIGA Genie 3" products at a monthly contract price. It was lowered to 9900 won. The service is provided at KRW 7700 per month for one-year contracts, KRW 5500 for two years, and KRW 4400 for three years. However, some AI speakers are sold without a monthly fee. The Google Nest Audio AI Bluetooth speaker is currently on sale for USD 120.

The security-related matters are very sensitive and refer to cases in which other people can obtain a user's personal information through voice recognition and use it incorrectly. The vulnerability of door opening or gas valve opening through malfunctioning artificial intelligence speakers has been reported in the media. When using AI speakers, commands can also be issued remotely. For this reason, it is possible to control the Internet of Things by hacking artificial intelligence speakers. The current level of artificial intelligence speakers is very limitedly connected to the Internet of Things, but when all things are connected, it can create a bigger problem [24–27]. Despite these security risks, we want to analyze whether people will feel the value of using them. In addition, we want to find out whether it affects the intention to continue using the product.

The purpose of this study is to measure user perception of information reliability, which is the last antecedent variable that can negatively affect the perceived value of artificial intelligence speakers. People who have used artificial intelligence speakers said that when asking for the requested information, it was difficult to receive an answer that correctly corresponded to the question asked. Simple questions could be answered, but complex questions were often not answered by the AI speaker.

Especially when asked about history, science, and mathematics, it is often not possible to receive an answer. This researcher even made a simple calculation request for some artificial intelligence speakers. As a result, the AI speaker was often unable to answer. For example, when asked for the answer of 2 + 2, among the simplest of mathematical questions, half of the 11 artificial speakers who answered four to the first answer did not answer. I do not know whether this is a problem with speech recognition or an error of the artificial intelligence algorithm inside, but it seems that it is a simple question that

humans think needs redefinition. Recently, the European Union (EU) has been making the following efforts to verify the reliability of artificial intelligence as well as the reliability of the information, including artificial intelligence speakers. A certification system is being implemented to evaluate the reliability of products and services and the appropriateness of the management system for the AI legislation related to "requirements for high-risk AI systems". According to a March 2022 press release, government ministries and related research institutes are planning to conduct AI reliability verification in Korea [27].

### 2.4. Endogenous Variable

Perceived value is a concept that asks if there is effort and savings in time and cost using artificial intelligence speakers. Through the anthropomorphism, intimacy, reliability, and perceived enjoyment provided by the AI speaker presented in CASA's paradigm, we will check whether the user feels the value of saving time, money, and effort and satisfying the expected effect [28–30]. The purpose of this study is to find and suggest improvement factors that affect perceived values through causality analysis. Recommendation intention is possible when there is a positive perception of perceived value. The reason for the recommendation is the usefulness and convenience of information retrieval and the purchase of everyday products in non-face-to-face situations [30].

It is the operating time of the peripheral device, and it is a case where an answer can be given to complex inferences and calculations. The purpose of this study is to find out which of the factors of convenience, usefulness, pleasure, and anthropomorphism that are expected to affect the perceived value presented in this study will give a positive factor. In addition, in the point of improvement, we try to find out which factors among technical difficulties, cost burden, security threats, and information reliability are the factors that urgently need improvement.

### 2.5. Hypothesis Setting

**H1.** *The convenience of the artificial intelligence speaker will have a positive (+) effect on the perceived value.*

**H2.** *The usefulness of artificial intelligence speakers will have a positive (+) effect on perceived value.*

**H3.** *The pleasure of the artificial intelligence speaker will have a positive (+) effect on the perceived value.*

**H4.** *The perceived anthropomorphism of the artificial intelligence speaker will have a positive (+) effect on the perceived value.*

**H5.** *The technical difficulties of the artificial intelligence speaker will negatively (−) affect the perceived value.*

**H6.** *The cost burden of artificial intelligence speakers will negatively (−) affect the perceived value.*

**H7.** *The security threat of artificial intelligence speakers will negatively (−) affect the perceived value.*

**H8.** *Information unreliability of artificial intelligence speakers will negatively (−) affect perceived value.*

**H9.** *The perceived value of the artificial intelligence speaker will have a positive (+) effect on the recommendation intention.*

### 3. Materials and Methods

#### 3.1. Materials

The analysis data used in this study were directly collected using an online survey tool, and the perception of the use of artificial intelligence speakers was investigated. For the question items, 11 items were composed to identify the characteristics of the respondent, and 8 constituent concepts were composed. Constructs were set for ease of use, usefulness, enjoyment, technical difficulty, the burden of cost, security risk, perceived value, and recommendation intention, and five sub-measurement variables were composed for each. A total of 40 queries were composed by configuring 5 measurement variables

in 8 constructs. The collection period was from May 1st to May 30th, regardless of age, gender, and nationality.

The total number of the collected statistics was 160, but 2 were removed because they could not complete the query, and 2 were removed by consistently checking the perceived value and recommendation intention with 5 points. Finally, 156 responses were collected. The subjects were those who currently used domestic and foreign artificial intelligence speakers at least once. In addition, many users were asked about the purpose of use and how many hours a day, on average, they used the speakers. In order to understand the usage characteristics of the respondents, the method of using artificial intelligence speakers as a direct purchase, additional service, or rental method under a telecommunication company contract was investigated. The level of technology that is usually recognized while using artificial intelligence speakers was investigated. In this regard, we inquired about the items that need to be urgently supplemented. Finally, the future appearance of the artificial speaker 10 years later was selected.

## 3.2. Methods

This study was conducted with people who had experience using artificial intelligence speakers regardless of brand. The survey items were constructed using a Korean Naver form, such as a Google survey, and an online survey was conducted. In order to analyze the characteristics of the users, age, gender, types, and methods of using artificial intelligence speakers and the average time and frequency of use per day were investigated. The main purpose of using artificial intelligence speakers and the current level of technology were asked. In addition, questions were asked about things that require urgent improvement based on user experience and the future in the next 10 years. Frequency analysis, which is a descriptive statistical analysis, will be performed on user characteristics. The questionnaire composition was measured by applying the Value-Based Acceptance Model (VAM) and dividing the antecedent variables affecting the perceived value into the positive and negative factors. The positive factors were convenience, usefulness, playfulness, and anthropomorphism. The negative factors were set as technical difficulties, cost burden, and security risks.In particular, considering that it has been more than 10 years since artificial intelligence speakers were released on the market, we set up a hypothesis that there would be an anthropomorphism property as the technology has advanced. In the analysis method, factor analysis and Cronbach's alpha analysis were performed to suggest the statistical significance of the validity and reliability of the questionnaire composition. Regression analysis was performed to analyze the influence of the positive and negative antecedent variables on the perceived value.

## 4. Results

The resulting analysis derived improvements based on the user's perception of the artificial intelligence speaker. First, in order to understand the usage status, the frequency analysis of the characteristics of the respondents was attempted. For the reliability of the respondents to the questionnaire, we wanted to check whether it is 0.7 or more through Cronbach analysis [29]. Correlation analysis was performed for the discriminant validity between the constructs [30]. Finally, through regression analysis, we determined which of the positive and negative factors for the perceived value is the improvement factor [30].

## 4.1. Research Model

In the research model of Figure 2, Ease of Use, Usefulness, Enjoyment, and Perceived Anthropomorphism were set as the Benefit factors affecting the Perceived value. Here, Perceived Anthropomorphism is a question of whether it functions similarly to humans. In fact, if you look at the AI speakers available on the market, Amazon has had a long-serving system called Alexa since November 2014. As of 2022, eight years later, people will be expecting technological advances. The preceding variables were set as to whether the artificial intelligence speakers are similar to humans due to technological advances. In

addition, regarding technical difficulties, many respondents who participated in this study complained of a lot of inconvenience in voice recognition. We want to check if there is a burden on the monthly billing amount and the cost of updating and replacing. The purpose of this study is to understand the perception of security risks in the process of operating peripheral devices through artificial intelligence speakers.

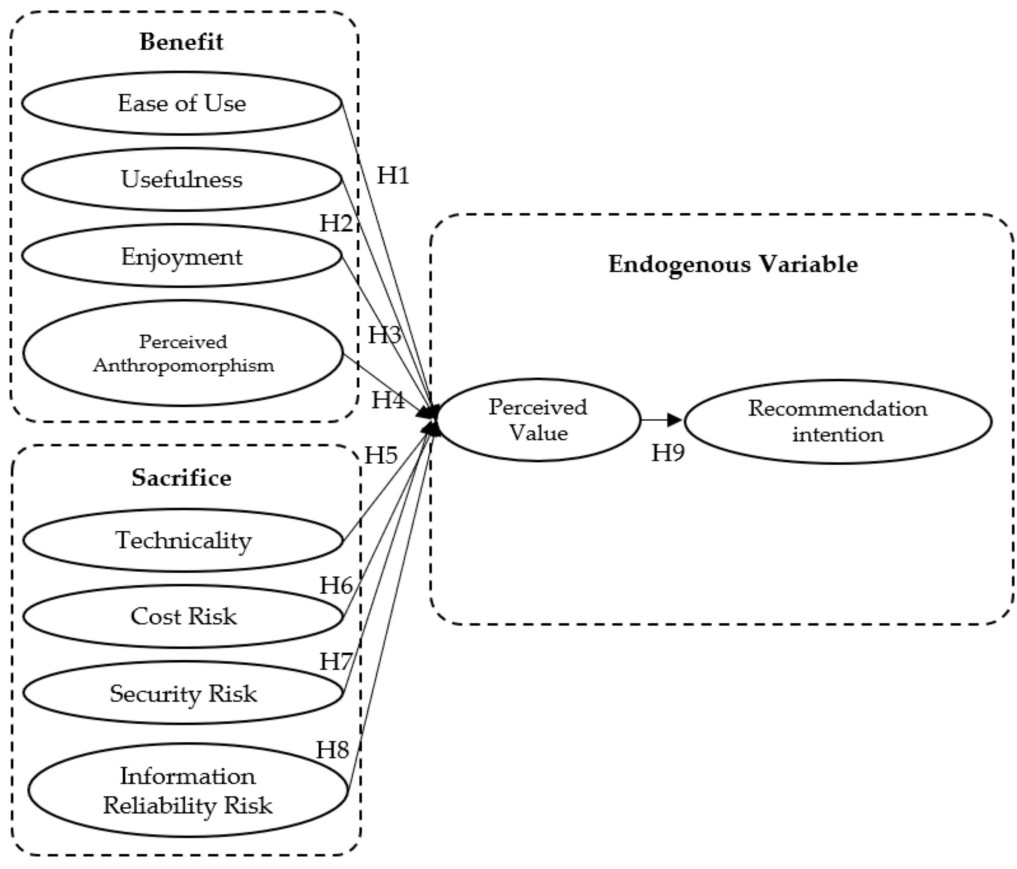

**Figure 2.** AI speaker-VAM research model [31,32].

Lastly, we want to check what kind of perception we have about the risk of information reliability. Before starting this study, we asked the participants if they could receive the information they wanted if they asked the AI speaker to search for information. However, most of the respondents said they were very disappointed. However, we cannot generalize with a small sample, so we want to find out whether the majority agree with that perception. In summary, at a time when there are more and more users, we would like to suggest which points need urgent improvement.

### 4.2. Analysis of the Characteristics of Respondents

The results of analyzing the characteristics of respondents are summarized in Table 2 below. The total number of respondents was 156, and among the respondents, the age of 20 or more and less than 30 was 41.7%, 48.7% were male, and 51.3% were female. As for the AI speakers used, KT (South Korea, KT Giga Gnie) was the most represented, with 25.0%, and Apple's HomePod was the most used, with 16.7% for overseas products. The average use time was less than 1 h, accounting for 80.1%, and the main purpose of use was information search (34.0%) and listening to music (33.0%). In addition, in terms of usage, the rate of contracting and using an additional service from a telecommunication company accounted for 59.0%. The average number of times a day was used was less than one time, 51.3%, and more than one time but less than five times, 41.7%. Through this study, we want to find out why the frequency of use is very low and the use time is less than 1 h.

**Table 2.** Result of analyzing the characteristics of respondents.

| Measured Variable | Frequency | Percent | Measured Variable | Frequency | Percent |
|---|---|---|---|---|---|
| 10 to 20 years old | 3 | 1.9 | Used less than 1 h | 125 | 80.1 |
| Over 20 to under 30 | 65 | 41.7 | Used for more than 1 h and less than 5 h | 29 | 18.6 |
| 30's or older–under 40's | 17 | 10.9 | Used for more than 5 h | 2 | 1.3 |
| Over 40 to under 50 | 35 | 22.4 | Total | 156 | 100.0 |
| over 50 years old | 36 | 23.1 | Conversation | 7 | 4.5 |
| Total | 156 | 100.0 | Listening to music | 52 | 33.3 |
| Male | 76 | 48.7 | Schedule and alarm Function | 8 | 5.1 |
| Female | 80 | 51.3 | Information retrieval | 53 | 34.0 |
| Totlal | 156 | 100.0 | Peripheral operation | 36 | 23.1 |
| KT (Korea, Giga Genie) | 39 | 25.0 | Total | 156 | 100.0 |
| LG Electronics (Korea, Pink Hub) | 8 | 5.1 | Rent to use | 9 | 5.8 |
| SKT (Korea, Nugu) | 25 | 16.0 | Buy to use | 55 | 35.3 |
| Google (USA, Home) | 11 | 7.1 | Additional service by telecommunication company contract | 92 | 59.0 |
| Naver (Korea, Wave) | 8 | 5.1 | Total | 156 | 100.0 |
| Samsung Electronics (Korea, Galaxy Home) | 24 | 15.4 | Used less than once | 80 | 51.3 |
| Xiaomi (China, Mi AI) | 7 | 4.5 | Used more than 1 time~less than 5 times | 65 | 41.7 |
| Alibaba (China, Tmall Genie) | 1 | 0.6 | Used more than 5 times | 11 | 7.1 |
| Apple (USA, HomePod) | 26 | 16.7 | Total | 156 | 100.0 |
| Kakao (Korea, Kakao Money) | 7 | 4.5 | | | |
| Total | 156 | 100.0 | | | |

Note: Analysis of usage characteristics.

Table 3 investigates the future prospects for technological level improvement for artificial intelligence speakers, where improvement is expected, and to what extent overall development can be achieved. Of the respondents, 49.4% of the respondents answered that their current AI technology level was at the initial level, 46.2% answered that they were at the intermediate level, and only 4.5% of those who answered that it was the advanced level. In addition, 57.1% answered the question of the reliability of information as an urgent matter to be improved, and 39.1% answered the need to improve technical problems such as voice recognition and reasoning. As a result of asking about the prospects of artificial intelligence speakers in 10 years, 48.7% said they would learn, reason, speak and act similarly to humans, and 42.9% said they would provide reliable information by applying surrounding IoT technology and big data.

*4.3. Reliability Analysis*

Table 4 analyzes the reliability of respondents for each construct. Reliability analysis was performed with SPSS version 20 (Seoul, Korea) to confirm the reliability of the questionnaire response, and it was confirmed that the reliability of the response was secured with a Chronbach $\alpha$ value of 0.7 or higher [33]. The minimum value of the Chronbach $\alpha$ value for the measurement variables of each component is 0.713 for information reliability, and the maximum value for the security threat is 0.927, so it can be said that the response is reliable.

**Table 3.** Analysis result of technology level, improvement items, and perception of the future.

| Outlook Element | Measured Variable | Frequency | Percent |
|---|---|---|---|
| Technology Level | Advanced level: The level at which you learn and talk similarly to humans. | 7 | 4.5 |
| | Intermediate level: Infer information and learn patterns to present analyzed information. | 72 | 46.2 |
| | Initial level: The level of information retrieval without reasoning and pattern learning. | 77 | 49.4 |
| | Total | 156 | 100.0 |
| Factors needing improvement | Technical functions: speech recognition technology, reasoning, and the ability to learn. | 61 | 39.1 |
| | Aesthetic function: Not a simple speaker shape, but a human or animal shape function. | 6 | 3.8 |
| | Information Reliability A function of filtering reliable and reliable information rather than simply collecting and answering surrounding information. | 89 | 57.1 |
| | Total | 156 | 100.0 |
| Preception of the future | In the next 10 years: AI speakers will provide reliable information with the development of big data and surrounding IoT technologies. | 67 | 42.9 |
| | In the next decade: AI speakers will reason and learn similarly to humans, speaking and acting similarly to humans. | 76 | 48.7 |
| | 10 years from now: I don't think there will be much of a difference in terms of the pace of development of artificial intelligence speakers. | 13 | 8.3 |
| | Total | 156 | 100.0 |

Note: Present and future level analysis result of artificial intelligence speaker.

**Table 4.** Reliability analysis results for constructs.

| Constructs | * Number of Sub-Factors of the Construct | Cronbach $\alpha \geq 0.70$ |
|---|---|---|
| Convenience | 5 | 0.847 |
| Usefulness | 5 | 0774 |
| Enjoyment | 5 | 0.837 |
| Anthropomorphism | 5 | 0.891 |
| Technical difficulties | 5 | 0.791 |
| Burden of expenses | 5 | 0.878 |
| Security threats | 5 | 0.927 |
| Information reliability | 5 | 0.713 |
| Perceived value | 5 | 0.819 |
| Recommendation intention | 5 | 0.832 |

* Refer to Appendix A for detailed sub-items of the constituent concept.

## 4.4. Correlation Analysis

To test the discriminant validity between the constructs, peer line correlation coefficient analysis was performed using SPSS. If there is 0.8 or more between the constructs, it is evaluated that the discriminant validity is not secured. In Table 5, there is no value greater than 0.8, so it can be said that there is discriminant validity between constructs. In addition, the significance probabilities of 0.05 and 0.01 levels were evaluated and marked with an asterisk. Negative values among the values indicate negative factors among the antecedent variables affecting the perceived value.

**Table 5.** Correlation analysis result.

| Constructs | 1 | 2 | 3 | 4 | 5 | 6 | 7 | 8 | 9 | 10 |
|---|---|---|---|---|---|---|---|---|---|---|
| Convenience | 1.00 | | | | | | | | | |
| Usefulness | 0.72 ** | 1.00 | | | | | | | | |
| Enjoyment | 0.65 ** | 0.71 ** | 1.00 | | | | | | | |
| Anthropomorphism | 0.44 ** | 0.65 ** | 0.55 ** | 1.00 | | | | | | |
| Technical difficulties | −0.03 | 0.01 | 0.08 | −0.14 | 1.00 | | | | | |
| Burden of expenses | 0.07 | 0.20 ** | 0.24 ** | 0.20 * | 0.27 ** | 1.00 | | | | |
| Security threats | 0.08 | 0.14 | 0.19 ** | 0.10 | 0.31 ** | 0.41 ** | 1.00 | | | |
| Information reliability | 0.06 | 0.13 | 0.14 | 0.09 | 0.52 ** | 0.29 ** | 0.27 ** | 1.00 | | |
| Perceived value | 0.49 ** | 0.59 ** | 0.62 ** | 0.39 ** | 0.11 | 0.21 ** | 0.40 ** | 0.20 ** | 1.00 | |
| Recommendation intention | 0.40 ** | 0.45 ** | 0.52 ** | 0.40 ** | 0.01 | 0.14 | 0.31 ** | 0.13 | 0.66 ** | 1.00 |

** Correlation is significant at the 0.01 level (2-tailed). * Correlation is significant at the 0.05 level (2-tailed).

## 4.5. Regression Analysis Result

Among the antecedent variables affecting the perceived value, usefulness (H2) and pleasure (H3) were found to have a positive effect. However, convenience (H1) and anthropomorphism (H4) were found to be improved. Respondents disagreed with the perception that artificial intelligence speakers came to the market 15 years ago and that technological progress would have improved to a level similar to that of humans. Even an artificial intelligence speaker that does not have this function will be hard to escape from the perception that it is a speaker with built-in automation. In addition, improvements are needed in the convenience of operating voice recognition and other devices or finding various information suitable for me, operating the device remotely, and reserving a restaurant. Among the functions of artificial intelligence speakers on the market, there is a shopping function, but it was found that these main functions did not work. Convenience is a very sensitive factor for the socially disadvantaged or for use by the elderly.

Table 5 confirms whether each construct has discriminant validity. In Table 5, if there is a value of 0.8 or more, there is a problem with multicollinearity [34,35], and it can be said that the construct concept has no discriminant power. In this case, the construct should be deleted in the regression analysis. However, the values presented as a result in Table 5 are a minimum of -0.03 and a maximum of 0.72, and each component is a discriminated concept. Therefore, there is no construct to be deleted.

Table 6 shows the regression analysis of the constructs with discriminant validity. The criteria for evaluating the results of regression analysis in Table 6 are as follows. Durbin-Watson is a criterion for testing the independence of the residuals when linear regression analysis or multiple regression analysis is performed. Durbin–Watson's value ranges from 0 to 4, and the closer to 2, the less autocorrelation [34]. In multicollinearity, it is judged that there is no problem when the value between the independent variable and the dependent variable is less than 10(VIF < 10) [35]. That is, it means a distinct construct concept. The criterion for statistical significance of accepting or rejecting the hypothesis in Table 6 is that the absolute value of the two-tailed test criterion t exceeds 1.96 and $p < 0.05$ [36].

That is, it is judged that the closer to 2, the less autocorrelation exists. R square is the square of the correlation coefficient between variables (items), and it is interpreted to be meaningful if it is 0.6 or more in academia and the 0.3 or more in marketing research

practice [37]. The meaning of this number refers to the degree to which the cause-and-effect variables are explained. Sig. of ANOVA tells whether the regression equation itself is significant [38]. That is, when Sig. is $p < 0.05$, it can be said that the regression equation is significant after analysis is performed. Table 6 satisfy the criteria for statistical significance.

**Table 6.** Results of regression analysis of antecedent variables affecting perceived value.

| Model | Unstandardized Coefficients | | Standardized Coefficients | t | Sig. | Collinearity Statistics | |
| --- | --- | --- | --- | --- | --- | --- | --- |
| | B | Std. Error | Beta | | | Tolerance | VIF |
| Constant | 0.845 | 0.309 | | 20.731 | 0.007 | | |
| (H1) Convenience | 0.053 | 0.073 | 0.066 | 0.729 | 0.467 | 0.413 | 2.424 |
| (H2) Usefulness | 0.233 | 0.110 | 0.251 | 20.113 | 0.036 | 0.237 | 4.219 |
| (H3) Enjoyment | 0.278 | 0.080 | 0.334 | 30.477 | 0.001 | 0.362 | 2.760 |
| (H4) Anthropomorphism | −0.013 | 0.061 | −0.017 | −0.217 | 0.828 | 0.531 | 1.882 |
| (H5) Technical difficulties | −0.048 | 0.081 | −0.043 | −0.590 | 0.556 | 0.639 | 1.565 |
| (H6) Burden of expenses | −0.047 | 0.055 | −0.057 | −0.854 | 0.395 | 0.753 | 1.328 |
| (H7) Security threats | 0.265 | 0.054 | 0.322 | 40.909 | 0.000 | 0.779 | 1.284 |
| (H8) Information reliability | 0.078 | 0.081 | 0.067 | 0.961 | 0.338 | 0.689 | 1.452 |
| **Durbin Watson** = 2.030, **R Square** = 0.507, **ANOVA(Sig.)** = 0.000 | | | | | | | |

Dependent Variable: Perceived Value.

The results of the regression analysis in Table 6 are as follows. The antecedent variable Convenience (H1), which is expected to have a positive (+) effect on the Perceived value, was rejected. Usefulness (H2) and Enjoyment (H3) were adopted. However, Anthropomorphism (H4) was rejected. Among the negative antecedent variables, the hypothesis that Technical difficulties (H5) and Burden of expenses (H6) would negatively affect the Perceived value was rejected. However, the hypothesis that security threats (H7) will negatively affect Perceived Value is adopted, so the security problem of artificial intelligence speakers in the future is a factor to be resolved. The hypothesis that information reliability (H8) will also have a negative (-) effect on the Perceived Value was adopted, indicating that the reliability of information provision needs to be supplemented.

Technical difficulty (H5) was rejected as an antecedent variable that could negatively affect perceived value, and cost burden (H6) and reliability of information (H8) were all rejected. In other words, it can be seen that the use of artificial intelligence speakers is not an improvement. However, it was adopted that the threat of security (H7) would have a negative impact. In other words, it can be said that improvement is necessary. First of all, the opposite results were derived from the facts that could be known through user interviews prior to this study. In the pre-interview, several people said that they recognized that there were many things that could not be answered or given a fixed answer as to the reliability of the information. It was found that there is no burden in paying the cost by dividing it into a monthly fee as an additional service of a telecommunication company rather than as a direct purchase. In terms of technical difficulties, it is somewhat inconvenient, but it can be said that it is satisfactory to search for desired content, such as voice recognition. However, there are still no technical difficulties, but it is not at the level to deliver convenience. In particular, with regard to security, it can be seen that personal information transmitted through the network and the connection between peripheral devices can be hacked, and improvements are needed for things that can cause malfunctions.

Finally, in Table 7, the hypothesis (H9) was adopted that perceived value would have a positive effect on recommendation intention. In other words, it is concluded that the currently used artificial intelligence speaker provides usefulness and enjoyment, has no technical difficulties and costs, and has reliable information. However, it can be seen that security threats need improvement.

**Table 7.** Results of analysis of the effect of perceived value on recommendation intention.

| Model | Unstandardized Coefficients | | Standardized Coefficients | t | Sig. | Collinearity Statistics | |
|---|---|---|---|---|---|---|---|
| | B | Std. Error | Beta | | | Tolerance | VIF |
| Constant | 0.997 | 0.224 | | 40.460 | 0.000 | | |
| (H9) Perceived value | 0.698 | 0.064 | 0.658 | 100.847 | 0.000 | 1.000 | 1.000 |
| **Durbin Watson** = 1.815, **R Square** = 0.433, **ANOVA(Sig.)** = 0.000 | | | | | | | |

Dependent Variable: Recommendation intention.

Figure 3 shows the results of testing the causal relationship by applying the VAM model. If the results of the antecedent variables that affect the perceived value of the artificial intelligence speakers are summarized as a value-based acceptance model, it can be presented as shown in Figure 3.

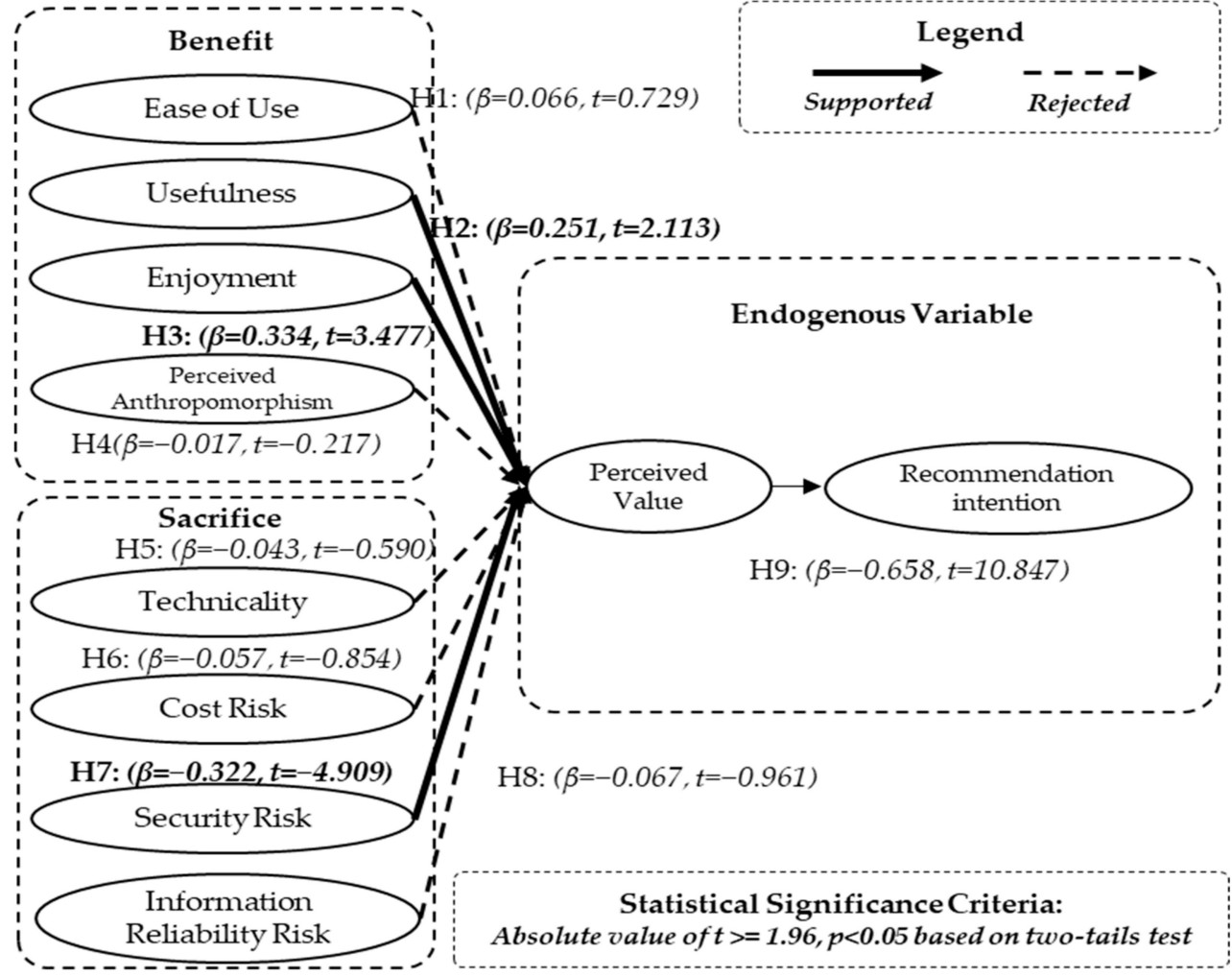

**Figure 3.** VAM applied artificial intelligence speaker regression analysis result.

Table 8 is the result of analyzing the effect of the preceding variable on the Perceived value according to the purpose of use. First of all, it was found that users who use peripheral devices for the purpose of operating them are aware of the value of using them for convenience and have an intention to recommend them to others. Users who use it for information retrieval were found to use it because of its usefulness.

Table 8. Causal relationship analysis according to the purpose of use.

| Purpose of Use | Construct | β | t | p | Reust |
|---|---|---|---|---|---|
| Peripheral operation | Constant | - | −0.806 | 0.427 | - |
| | Convenience | 0.435 | 2.378 | 0.025 | Supported |
| | Usefulness | 0.258 | 1.053 | 0.302 | Rejected |
| | Enjoyment | 0.096 | 0.420 | 0.678 | Rejected |
| | Anthropomorphism | −0.038 | −0.229 | 0.821 | Rejected |
| | Technical difficulties | 0.180 | 1.094 | 0.284 | Rejected |
| | Burden of expenses | −0.101 | −0.576 | 0.570 | Rejected |
| | Security threats | 0.290 | 1.561 | 0.130 | Rejected |
| | Information reliability | 0.182 | 1.187 | 0.245 | Rejected |
| Perceived Value →Recommendation Intention | | 0.935 | 5.877 | 0.002 | Supported |
| Information retrieval | Constant | - | 1.498 | 0.141 | - |
| | Convenience | -0.161 | −0.944 | 0.350 | Rejected |
| | Usefulness | 0.483 | 2.394 | 0.021 | Supported |
| | Enjoyment | 0.214 | 1.295 | 0.202 | Rejected |
| | Anthropomorphism | 0.046 | 0.339 | 0.736 | Rejected |
| | Technical difficulties | 0.042 | 0.289 | 0.774 | Rejected |
| | Burden of expenses | −0.120 | −0.975 | 0.335 | Rejected |
| | Security threats | 0.410 | 3.345 | 0.002 | Supported |
| | Information reliability | −0.044 | −0.294 | 0.770 | Rejected |
| Perceived Value →Recommendation Intention | | 0.687 | 6.693 | 0.000 | Supported |
| Schedule and alarm function | Constant | - | −58.388 | 0.011 | - |
| | Convenience | 0.959 | 111.080 | 0.006 | Supported |
| | Usefulness | −0.654 | −41.867 | 0.015 | Supported |
| | Enjoyment | 0.931 | 75.624 | 0.008 | Supported |
| | Anthropomorphism | 0.250 | 33.545 | 0.019 | Supported |
| | Technical difficulties | 0.708 | 69.151 | 0.009 | Supported |
| | Burden of expenses | −0.334 | −33.805 | 0.019 | Supported |
| | Security threats | 0.310 | −31.201 | 0.018 | Supported |
| | Information reliability | 0.309 | −30.210 | 0.015 | Supported |
| Perceived Value →Recommendation Intention | | 0.744 | 2.728 | 0.034 | Supported |
| Listening to music | Constant | - | 2.918 | 0.006 | - |
| | Convenience | −0.027 | −0.183 | 0.855 | Rejected |
| | Usefulness | 0.256 | 1.132 | 0.264 | Rejected |
| | Enjoyment | 0.653 | 3.672 | 0.001 | Supported |
| | Anthropomorphism | −0.148 | −1.134 | 0.263 | Rejected |
| | Technical difficulties | −0.318 | −2.830 | 0.007 | Supported |
| | Burden of expenses | 0.046 | 0.420 | 0.677 | Rejected |
| | Security threats | 0.175 | 1.751 | 0.087 | Rejected |
| | Information reliability | 0.117 | 1.087 | 0.283 | Rejected |
| Perceived Value →Recommendation Intention | | 0.632 | 5.824 | 0.000 | Supported |
| Conversation | Constant | - | 1.490 | 0.275 | - |
| | Convenience | 1.245 | 0.555 | 0.634 | Rejected |
| | Usefulness | −1.767 | −0.561 | 0.631 | Rejected |
| | Enjoyment | 0.328 | 0.201 | 0.859 | Rejected |
| | Anthropomorphism | 0.805 | 0.403 | 0.726 | Rejected |
| | Technical difficulties | −0.710 | −0.301 | 0.801 | Rejected |
| | Burden of expenses | −0.510 | −0.204 | 0.701 | Rejected |
| | Security threats | −0.140 | −0.010 | 0.210 | Rejected |
| | Information reliability | −0.321 | −0.211 | 0.400 | Rejected |
| Perceived Value →Recommendation Intention | | 0.615 | 4.543 | 0.000 | Supported |

Dependent variable: Perceived Value.

However, it was found to have a negative effect on the perceived value as a security threat. It was found that convenience, usefulness, pleasure, and personification had a

positive effect on perceived value for those who use it as a schedule and alarm function. In addition, technical difficulties, cost burden, security threats, and reliability of information did not affect the perceived value. For those who used artificial intelligence speakers for the purpose of listening to music, pleasure had an effect on perceived value. However, perceived technical difficulties appeared to be of no value. It does not provide any perceived value for purposes of dialogue. In other words, the current artificial intelligence speaker is similar to an electronic clock, not an artificial intelligence speaker used for schedule and alarm functions.

Table 9 shows the results of analyzing the effect of the antecedent variable on the Perceived value according to age. As a result of analyzing the usage status of artificial intelligence speakers by age, the anthropomorphic function is recognized as a perceived value in teens and people in their twenties.

**Table 9.** Causal relationship analysis by age.

| Purpose of Use | *Construct* | *β* | *t* | *p* | *Result* |
|---|---|---|---|---|---|
| Teenagers to under 20s | Constant | - | 0.238 | 0.852 | - |
| | Convenience | 0.500 | 0.577 | 0.667 | Rejected |
| | Usefulness | 0.756 | 1.155 | 0.454 | Rejected |
| | Enjoyment | 0.866 | 1.732 | 0.333 | Rejected |
| | Anthropomorphism | 0.977 | 4.619 | 0.000 | Supported |
| | Technical difficulties | −0.933 | −8.660 | 0.023 | Supported |
| | Burden of expenses | −0.901 | −8.545 | 0.020 | Supported |
| | Security threats | −0.918 | −2.309 | 0.026 | Supported |
| | Information reliability | −0.397 | −1.768 | 0.078 | Rejected |
| Perceived Value ➔ Recommendation Intention | | 0.277 | 0.289 | 0.821 | Rejected |
| 20′s to 30′s | Constant | | 2.178 | 0.034 | - |
| | Convenience | 0.210 | 1.368 | 0.177 | Rejected |
| | Usefulness | −0.121 | −0.522 | 0.604 | Rejected |
| | Enjoyment | 0.451 | 2.394 | 0.020 | Supported |
| | Anthropomorphism | 0.037 | 0.269 | 0.789 | Rejected |
| | Technical difficulties | −0.036 | −0.312 | 0.756 | Rejected |
| | Burden of expenses | −0.094 | −0.861 | 0.393 | Rejected |
| | Security threats | 0.257 | 2.473 | 0.016 | Supported |
| | Information reliability | 0.292 | 2.516 | 0.015 | Supported |
| Perceived Value ➔ Recommendation Intention | | 0.715 | 8.186 | 0.000 | Supported |
| 30′s or older~ under 40′s | Constant | | −0.165 | 0.874 | - |
| | Convenience | 0.178 | 0.594 | 0.571 | Rejected |
| | Usefulness | 0.719 | 2.539 | 0.039 | Supported |
| | Enjoyment | 0.475 | 1.626 | 0.148 | Rejected |
| | Anthropomorphism | −0.805 | −2.831 | 0.025 | Supported |
| | Technical difficulties | −0.165 | −0.500 | 0.633 | Rejected |
| | Burden of expenses | 0.066 | 0.183 | 0.860 | Rejected |
| | Security threats | 0.180 | 0.504 | 0.629 | Rejected |
| | Information reliability | 0.118 | 0.428 | 0.682 | Rejected |
| Perceived Value ➔ Recommendation Intention | | 0.014 | 0.052 | 0.952 | Rejected |
| Over 40~ Under 50 | Constant | | −0.809 | 0.426 | - |
| | Convenience | −0.159 | −0.928 | 0.362 | Rejected |
| | Usefulness | 0.393 | 1.923 | 0.065 | Rejected |
| | Enjoyment | 0.401 | 2.586 | 0.016 | Supported |
| | Anthropomorphism | 0.125 | 0.843 | 0.407 | Rejected |
| | Technical difficulties | 0.122 | 0.764 | 0.452 | Rejected |
| | Burden of expenses | −0.002 | −0.016 | 0.987 | Rejected |
| | Security threats | 0.404 | 3.111 | 0.004 | Supported |
| | Information reliability | −0.116 | −0.675 | 0.506 | Rejected |

**Table 9.** *Cont.*

| Purpose of Use | Construct | β | t | p | Result |
|---|---|---|---|---|---|
| Perceived Value →Recommendation Intention | | 0.712 | 5.824 | 0.000 | Supported |
| over 50 | Constant | | 0.196 | 0.846 | |
| | Convenience | 0.083 | 0.349 | 0.730 | Rejected |
| | Usefulness | 0.150 | 0.547 | 0.589 | Rejected |
| | Enjoyment | 0.274 | 1.322 | 0.197 | Rejected |
| | Anthropomorphism | 0.364 | 1.999 | 0.056 | Rejected |
| | Technical difficulties | 0.048 | 0.309 | 0.760 | Rejected |
| | Burden of expenses | 0.219 | 1.392 | 0.175 | Rejected |
| | Security threats | 0.203 | 1.400 | 0.173 | Rejected |
| | Information reliability | −0.110 | −0.847 | 0.405 | Rejected |
| Perceived Value →Recommendation Intention | | 0.715 | 5.959 | 0.000 | Supported |

Dependent variable: Perceived Value.

However, they are feeling the technical difficulties, the burden of cost, and the threat to security. They also responded that they did not feel the perceived value and had no intention of recommending it. Those in their twenties and thirties responded that only pleasure felt perceived value. It was found that it did not give perceived value in terms of security threats and the reliability of information. This means that most of them are only used for listening to music. Those in their thirties and forties responded that they felt perceived value for usefulness.

In the end, I have no intention of recommending it. It was found that people in their forties and fifties use them only for pleasure, similar to those in their twenties and thirties. It was found that those in their fifties and over did not feel the perceived value of artificial intelligence speakers at all. Despite the continuous growth of the global artificial intelligence speaker market, artificial intelligence speakers are no longer. It should not be called an artificial intelligence speaker. It appears that the brand name is simply called that way, and it is an automated food recognition speaker that cannot feel the perceived value.

## 5. Discussion

Recently, various products and software containing artificial intelligence, such as artificial intelligence speakers, artificial intelligence home appliances, cars with built-in artificial intelligence, artificial intelligence robots, and artificial intelligence chatbots, have been launched on the market. However, it is questionable whether it is performing such a function enough to attach a special artificial intelligence function. What on Earth is the technological level of artificial intelligence that people are aware of? Before proceeding with this study, I had an honest conversation with a fellow professor who studies artificial intelligence. In their story, I was told that inferences through big data-based data learning support people's efficient decision-making.

However, it is said that a lot of development is still needed to develop algorithms that learn and reason by collecting, processing, and processing big data. If so, is it at a level that users can agree to with the words of professors who study artificial intelligence? In fact, as a result of asking a few users and their perception of the artificial intelligence speaker, they said that it does not seem to function enough to be described as artificial intelligence. In particular, it is said that most people do not understand when a sentence used by a person is used by voice recognition.

They even say that it is similar to an automated function, not artificial intelligence, to the extent that there is a guideline on how to speak to an AI speaker for voice recognition. Even worse, some say that it adjusts the channel by installing voice recognition on the radio. They say they are not just talking about AI speakers. Products released on the market with artificial intelligence are said to be more sensitive and natural to movement through sensors but are not considered to be artificial intelligence. So, we asked users to what extent they would have to be able to attach the functional term of artificial intelligence.

They say that when people use natural language, they should understand it well and not just develop an algorithm that finds an appropriate answer because there is a lot of data stored there. It has a function of reasoning when a certain query is made, so it is said that it is necessary to present not only the general various choices and alternatives but also the choices and alternatives considering the environment of the person making the inquiry. If improvement is suggested based on the results of this study, it is useful and enjoyable, but it is still inconvenient to use. In addition, it was found that it is difficult to expect a level similar to that of a person, such as anthropomorphism. The function of anthropomorphism is that, as mentioned above, it must have the aspect of thinking, speaking, analyzing, and emotionally similar to a human by collecting data and reasoning as a human would, and considering the environment. Although technical difficulties, the burden of cost, and the reliability of the information, which are antecedent variables that would have a negative impact on perceived value, had a positive effect on perceived value, some improvement was needed. Despite these improvements, it is concluded that the perceived value of people is recommendable to others. Putting this together, it can be summarized as follows. What people expect of artificial intelligence is that it does not simply provide automation functions. In relation to the results of this study, although we are still somewhat satisfied with the speaker with the functional name of artificial intelligence, it is ultimately concluded that it is not an artificial intelligence speaker.

## 6. Conclusions

It has been more than 15 years since artificial intelligence speakers were released. Products released on the market for technological evolution are as follows. It would be more accurate to say "AI-oriented speaker" rather than "AI speaker". Of course, typical manufactured products are clearly different from AI-oriented speakers that need to evolve. Based on the results of this study, it is not at the level where the function of artificial intelligence can be attached to any product. It is clear that the term artificial intelligence is no longer a term with a technical function but a marketing term that sells well in the market.

The current level of AI mainly listens to music and sometimes delivers the weather through voice recognition or sounds an alarm. Considering that it has been 15 years since artificial intelligence speakers were released, this is not to say that artificial intelligence has not progressed. Recently, in the case of artificial intelligence robots, they have made humanoids and introduced them through exhibitions and the media that are very similar to humans.

It is said that the smiling expression is the same as that of a human being, and it is said that it is at the level of not only answering questions but also asking questions to humans. Robots that provide this level of functionality are priced in the hundreds of millions of dollars. Do we want more than a few hundred million technology levels for artificial speakers? It does not matter whether the speakers we use in our daily life are actually artificial intelligence speakers or well-developed automated speakers.

However, the basic functions against threats to convenience and security need to be improved, and the function of anthropomorphism is not at the current expected level but needs improvement. In fact, even after 15 years have passed since it was launched on the market, users did not feel burdened by the burden of pricing and marketing policies, and the communication companies that developed it and the companies that developed the machine do not feel the pressure of choice. The reality is that it is not easy to obtain. In this study, the recognition level of users for artificial intelligence speakers was evaluated. In the future, user recognition of products prefixed with artificial intelligence will be conducted in future research. What is clear is that there are only AI-oriented products but no products with AI functions. In addition, it is necessary to present what is different from the existing ones in the description, promotion, and marketing of functions without exaggeration so as not to create misunderstandings in the formation of these users' expectations. As mentioned above, artificial intelligence-oriented speakers will have a positive effect on

forming expectations for users by revealing that certain functions have been added and are under development.

**Author Contributions:** Conceptualization, S.-J.Y. and M.-Y.K.; Data curation, S.-J.Y.; Formal analysis, S.-J.Y.; Methodology, S.-J.Y. and M.-Y.K.; Project administration, M.-Y.K.; Resources, M.-Y.K.; Writing—original draft, S.-J.Y.; Writing—review & editing, S.-J.Y. and M.-Y.K. All authors have read and agreed to the published version of the manuscript.

**Funding:** This research received no external funding.

**Institutional Review Board Statement:** Not applicable.

**Informed Consent Statement:** Not applicable.

**Data Availability Statement:** Not applicable.

**Conflicts of Interest:** The authors declare no conflict of interest.

## Appendix A

The table below is a sub-measurement variable for the constructs of Table 4. Reliability analysis results for the constructs. There are five sub-queries for each construct and a total of 65 queries.

**Table A1.** Questionnaire composition.

| Constructs | Measured Variable |
| --- | --- |
| Convenience | 1. When I use an artificial intelligence speaker, it is convenient to use it with voice recognition support. <br> 2. I can conveniently operate the phone, TV, etc. using the artificial intelligence speaker. <br> 3. It is convenient for me to search for various information through the AI speaker. <br> 4. It is convenient for me to operate remote devices through artificial intelligence speakers. <br> 5. It has become convenient for me to make reservations and pay using the AI speaker. |
| Usefulness | 1. I think that the search results through the AI speaker are useful content for me. <br> 2. I think it is useful to provide various contents such as music, news, life, and economy through artificial intelligence speakers. <br> 3. I am useful as a conversation partner with an AI speaker. <br> 4. I can deliver and order non-face-to-face through an artificial intelligence speaker, which is useful. <br> 5. I think that artificial intelligence speakers provide usefulness to the elderly living alone and the socially disadvantaged. |
| Enjoyment | 1. I enjoy listening to music in my daily work and life through the AI speaker <br> 2. I enjoy being able to receive the latest popular movie information through the artificial intelligence speaker. <br> 3. I enjoy talking to AI speakers. <br> 4. I find it interesting and enjoyable to operate peripherals through artificial intelligence speakers. <br> 5. I enjoy listening to the schedule, the weather, and today's attire through the AI speaker. |

**Table A1.** *Cont.*

| Constructs | Measured Variable |
|---|---|
| Anthropomorphism | 1. I feel like the AI speaker is acting on its own will.<br>2. I feel like I'm talking to a human when I'm talking to an AI speaker.<br>3. I think that AI speakers have a certain consciousness like humans by analyzing people's life patterns.<br>4. I seem to refer to the emotional part of the user when making requests to the AI speaker, such as music or movies.<br>5. I don't think AI speakers are sometimes machines. |
| Technical difficulties | 1. I do not understand well when I ask the AI speaker by voice.<br>2. When I ask the AI speaker to search for information, sometimes I can't give an appropriate answer.<br>3. I often cause malfunctions when I give work instructions to the AI speaker by voice.<br>4. When I ask complicated questions to the AI speaker, It tells them that it is an unsupported function.<br>5. When I use an artificial intelligence speaker, it seems that I do not answer by reasoning and learning, but only giving a fixed answer. |
| Burden of expenses | 1. I feel the burden of cost when using artificial intelligence speakers.<br>2. When I buy an AI speaker, I feel burdened with the cost.<br>3. I feel the burden of cost when using content through artificial intelligence speakers.<br>4. When I use an artificial intelligence speaker, I feel burdened by the cost of additional services for a monthly fee.<br>5. I feel burdened by the cost of repair and replacement when the AI speaker malfunctions |
| Security threats | 1. I think that if I use an artificial intelligence speaker, my information may be stolen by connecting various communication devices..<br>2. I think that when security is set using artificial intelligence speaker, it can threaten the security device through hacking.<br>3. I think there is a risk of collecting private conversations on AI speakers and sharing them with others.<br>4. I believe that payment information may be leaked when purchasing, ordering, or making a reservation through an artificial intelligence speaker.<br>5. I think that several devices may be at risk for security by leaving the security settings open when the artificial intelligence speaker operates peripheral devices |
| Information reliability | 1. There were times when I got completely unrelated answers to the information provided by the AI speaker.<br>2. I often couldn't answer the information provided by the artificial intelligence speaker.<br>3. When I asked a question to an AI speaker, there were times when I repeated the same unreliable answer.<br>4. I think that most of the time, AI speakers find and provide information that is not based on the Internet.<br>5. I think AI speakers only provide fixed answers. |

**Table A1.** *Cont.*

| Constructs | Measured Variable |
|---|---|
| Perceived value | 1. I believe that artificial intelligence speakers are worth using because they provide convenience, usability, and enjoyment.<br>2. I think that artificial intelligence speakers are worth using for various in formation and content provision.<br>3. I think that artificial intelligence speakers are worth using because they have the ability to help the socially disadvantaged in people's lives.<br>4. I think that the AI speaker is worth using by providing a variety of information not only with limited information, but with fused and inferred information.<br>5. I think AI speakers can relieve some of the emotional (loneliness, sadness, joy) part. |
| Recommendation intention | 1. I would like to recommend artificial intelligence speakers to anyone who adds convenient information search and quickness.<br>2. I would recommend artificial intelligence speakers, including the socially disadvantaged, even for the convenience of life.<br>3. I would recommend using an AI speaker if it is necessary to purchase, order, or reserve an AI speaker in a non-face-to-face environment.<br>4. I would recommend if you want to operate/control peripherals with one voice recognition of an artificial intelligence speaker.<br>5. I would recommend using an AI speaker if you want accuracy and immediacy in complex reasoning and calculations of the AI speaker. |

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
