# Peer review of "A Study on Deriving Improvements through User Recognition Analysis of Artificial Intelligence Speakers"

_applsci, doi:10.3390/app12199651_

Round 1

Reviewer 1 Report

The paper conducts a study on artificial intelligence speakers and collects views of the users to understand their perception. The paper discusses the positive and negative factors of using AI-based speakers. The research poses some interesting questions e.g. if these are AI-based or automated speakers. Some useful statistical analysis have been provided. 

There are, indeed, some good points which add value to this paper. However, it seems like, it is covering multiple aspects and losing the coherence. Consider, re-structuring the paper clearly stating what you attempted to achieve linking this through the methodology. Discuss a bit more about how different age groups and aptitude levels may have affecting survey findings.

Author Response

Dear reviewer. hello?
I have carefully considered what you have pointed out. In fact, by analyzing the purpose of use and age, we were able to clarify the substance called artificial intelligence more clearly. Thanks for the benefit.

The reflected part was corrected and the uploaded file page 14~16 causal relationship analysis was performed and detailed explanation was written.

Thank you for your good rating and review.

Reviewer 2 Report

In the paragraph 3.1 "Material" there is a repetition in the first sentence. It is advisable to read the contribution again.

Author Response

Dear reviewer. hello?
The repeated words in 3.1 that you pointed out have been deleted.
The reflected page is page 7, line 271.

Thank you for your good rating and review.

Reviewer 3 Report

The paper aims to analyze the causal relationship related to the use of artificial intelligence speakers by people, and the perception they have about these artifacts. The objective is to analyze if the users perceive the artificial intelligent speakers as intelligent artifacts or simple automata that elaborate data.

For their purpose, the authors refer to the Value-Based Acceptance Model (VAM).

In my personal opinion, the study seems interesting, and the provided results are completed.

However, some points should be clarified and detailed.

The authors claim that VAM is a unique model that can be used for their purpose.

It is a strong affirmation, and they should be demonstrated that.

For example, they could refer to the work at [1], where preliminary results about the role of inner speech in a robotic artifact (that could be intended as a speaker too) in anthropomorphism of the artifacts were presented.

The mentioned work shows another model for evaluating what they do.

The bibliographies related to the existing artificial speakers on the market should be extended, that is the authors should provide a deeper textual description of figure 1

In particular, in the introduction, it should be necessary to add some bibliographies related to the mentioned points describing the reasons why people do not perceive AI as AI.

Refuse occurs in row 231 (repetition of "the the").

I think that is important for the authors to refine the bibliography and some claims as mentioned above for making the paper more complete.

[1] Pipitone, A., Geraci, A., D'Amico, A., Seidita, V., & Chella, A. (2021). Robot's Inner Speech Effects on Trust and Anthropomorphic Cues in Human-Robot Cooperation. arXiv preprint arXiv:2109.09388.

Author Response

Dear reviewer. hello?
What you have pointed out is the following:
1. Explain why you used the VAM model.
  2nd page reflected line 62~82
2. Explain in detail about Figure 1 and expand it in the market area.
The reflected page is page 3, line 117~125
3. On line 231, the repetition
  Repeated words, misspellings, and grammar have been corrected for the entire thesis.
4. Need to explain important parts of the author suggested by the reviewer
  Lines 147 to 156 on page 4 reflected.
  References have been added to Attachment No. 17.

Round 2

Reviewer 1 Report

The paper reads way better than the earlier version. Consider being more consistent in referencing your sources e.g. somewhere you uses Harvard referencing format stating authors and publication dates (author(2010)) and other places you use numbers ([2]). Stick to the required format for the paper.